# Comprehensive analysis of the functional impact of single nucleotide variants of human *CHEK2*

**Claire E. McCarthy-Leo**[1], **George S. Brush**[2], **Roger Pique-Regi**[1,3], **Francesca Luca**[1,3¤], **Michael A. Tainsky**[1,2], **Russell L. Finley, Jr**[1]*

**1** Center for Molecular Medicine and Genetics, Wayne State University School of Medicine, Detroit, Michigan, United States of America, **2** Department of Oncology, Molecular Therapeutics Program, Barbara Ann Karmanos Cancer Institute, Wayne State University School of Medicine, Detroit, Michigan, United States of America, **3** Department of Obstetrics and Gynecology, Wayne State University School of Medicine, Detroit, Michigan, United States of America

¤ Current address: Department of Human Genetics, University of Chicago, Chicago, Illinois, United States of America

* rfinley@wayne.edu

**Data Availability Statement:** The authors confirm that all data underlying the findings are fully available without restriction. All relevant data are

## Abstract

Loss of function mutations in the checkpoint kinase gene *CHEK2* are associated with increased risk of breast and other cancers. Most of the 3,188 unique amino acid changes that can result from non-synonymous single nucleotide variants (SNVs) of *CHEK2*, however, have not been tested for their impact on the function of the *CHEK2*-enocded protein (CHK2). One successful approach to testing the function of variants has been to test for their ability to complement mutations in the yeast ortholog of *CHEK2*, *RAD53*. This approach has been used to provide functional information on over 100 *CHEK2* SNVs and the results align with functional assays in human cells and known pathogenicity. Here we tested all but two of the 4,887 possible SNVs in the *CHEK2* open reading frame for their ability to complement *RAD53* mutants using a high throughput technique of deep mutational scanning (DMS). Among the non-synonymous changes, 770 were damaging to protein function while 2,417 were tolerated. The results correlate well with previous structure and function data and provide a first or additional functional assay for all the variants of uncertain significance identified in clinical databases. Combined, this approach can be used to help predict the pathogenicity of *CHEK2* variants of uncertain significance that are found in susceptibility screening and could be applied to other cancer risk genes.

## Author summary

Variations in gene sequences account for many of the phenotypic differences between individuals, including different susceptibilities to diseases like cancer. Individuals who inherit specific variants in the *CHEK2* gene, for example, have increased risk of developing breast cancer and other cancers. These cancer-associated variants all abolish the normal activity of the *CHEK2*-encoded protein, a protein kinase called Chk2. Most of the *CHEK2*

within the paper and its Supporting Information files. Raw data used for producing all graphs and figures are in Supplemental Tables 1, 2, and 3.

**Funding:** This work was supported by a grant from the Wayne State University Center for Molecular Medicine and Genetics (to RLF), the Barbara and Fred Erb Endowed Chair in Cancer Genetics to MAT, and Rumble Fellowship from the Graduate School of Wayne State University (CML). The funders had no role in study design, data collection and analysis, decision to publish, or preparation of the manuscript.

**Competing interests:** The authors have declared that no competing interests exist.

variants found in individuals, however, have an unknown effect on the function of the Chk2 protein, and therefore an unknown potential role in cancer. This is because it is cumbersome to individually make and test each newly identified variant to see whether it affects protein function. Here we use an efficient version of a technique called deep mutational scanning to test nearly all possible *CHEK2* single nucleotide variants for their impact on Chk2 protein function. Analyses of the resulting data indicate that it will be useful for developing models to predict the cancer risk in individuals who inherit any *CHEK2* variants. This study also demonstrates a viable efficient approach to study variants of other genes, including other cancer related genes.

## Introduction

Cell cycle checkpoint kinase 2 (CHK2) is a conserved protein involved in cell cycle control and DNA damage responses. First discovered in 1998 as the mammalian ortholog of Rad53 in *Saccharomyces cerevisiae*, CHK2 has established roles in DNA double strand break repair, DNA damage-induced apoptosis and cell cycle arrest [1]. CHK2 is a 543 amino acid nuclear serine/threonine protein kinase that contains three distinct functional domains; an SQ/TQ cluster domain (SCD), a forkhead-associated (FHA) domain and a kinase domain. When DNA damage occurs, activated ATM phosphorylates CHK2 at T68 [2–4]. This phosphorylation induces dimerization of CHK2 monomers through the binding of the phosphorylated SCD of one CHK2 monomer with the FHA domain of another, initiating autophosphorylation within the kinase domain (S260, T420), the activation loop (T383, T387), and the C-terminus (S516) [5–7]. Activated CHK2 monomers spread the DNA damage signal by phosphorylating downstream protein targets to halt the cell cycle, recruit various proteins to the site of DNA damage, and initiate apoptosis when necessary [1,8–10].

*CHEK2* is a moderately penetrant gene that has been studied for its association with cancer predisposition [11]. Women with the heterozygous founder mutation *CHEK2* c.1100delC, which leads to a protein truncation in the kinase domain, have a 2-3-fold increased risk of developing breast cancer [12–14]. This pathogenic variant has also been associated with increased risk for male breast cancer, gastric, testicular, prostate, and thyroid cancers [15–20]. Additional *CHEK2* variants that impair protein function have been identified in the germline of individuals with cancer and are associated with the development of cancer [21–27]. Standards for the clinical classification of sequence variants established by the American College of Medical Genetics and Genomics (ACMG) use multiple pieces of evidence (family history, allele frequency, segregation and functional data, etc.) to classify sequence variants as either pathogenic (associated with increased cancer risk) or benign [28]. Most variants detected in susceptibility screening, however, lack supporting evidence by statistically significant population data to classify them as either pathogenic or benign, resulting in their classification as variants of uncertain significance (VUS) [28]. One solution to this problem is to test individual variants using assays for protein function to determine whether each variant is damaging to protein function. Functional analysis can provide supporting evidence for clinical classification of rare variants that otherwise could not be supported by population and family data [11,28].

Several different functional assays have been used to test individual and small sets of CHK2 missense variants identified in patients, including tests of protein stability, kinase activity toward various substrates, and activation by DNA damaging agents [22,23,29–32]. Another method to assay the function of CHK2 variants is to test their ability to complement a yeast *RAD53* loss-of-function mutant (*rad53*). Human *CHEK2* was cloned and first tested for its ability to complement yeast *rad53* in a complementation growth assay in 1998 [1]. This study

also demonstrated that the CHK2 kinase defective mutant, D347A, failed to complement *rad53*. Yeast rad53 complementation has been used as a method for protein function assessment on several other *CHEK2* pathogenic and likely-pathogenic variants. For example, the *CHEK2* founder mutation, c.1100delC was tested for complementation of yeast *rad53* mutants and no complementation was observed [24]. The same study also showed that a *CHEK2* missense variant leading to S428F within the kinase domain does not complement, while a missense variant leading to P85L in the SCD has wildtype-like activity. Additionally, *rad53* complementation assays using the DNA damaging agent, methyl methanesulfonate (MMS) have been performed to evaluate the functional impact of over 100 germline missense variants detected in familial breast cancer patients or selected from the ClinVar database [33–35]. Over 2,700 *CHEK2* missense variants, however, have not been tested in a functional assay, including most of the VUS in the ClinVar database.

To test the ability of all possible *CHEK2* coding SNVs to complement a *rad53* mutant, we conducted a version of deep mutational scanning (DMS) [36,37]. This technique, also known as a multiplex assay of variant effects (MAVE) [38], has become a valuable tool to evaluate the impact of genetic variants on protein function and has been used in several studies to characterize human protein variants based on their function in yeast [39–43]. To carry out a DMS screen. we made a comprehensive library of *CHEK2* coding variants using error-prone PCR, screened the library *en masse* for *rad53* complementation, then used next generation sequencing (NGS) to identify complementors in the selected yeast. This resulted in testing 4,885 of the 4,887 possible *CHEK2* SNVs, including all but one of the 3,913 possible codon changes (stop, synonymous, or non-synonymous). This large-scale functional assay of CHK2 variants evaluated the impact of almost every possible SNV in the open reading frame of *CHEK2*, providing data to aid in the classification of *CHEK2* VUS.

## Results

### A Screen for Variants That Affect CHK2 Function

To construct a library of all possible *CHEK2* SNVs, we used error-prone PCR (EP-PCR) on the *CHEK2* open reading frame (ORF) and subcloned the mutagenized ORF into a yeast expression vector (Methods). Subsequent next generation sequencing of the library after introduction into yeast revealed an average SNV frequency of 0.86 (SD = 0.46) per 1,000 bp, or 1.4 SNV per *CHEK2* ORF. The distribution of different variants is very similar to that reported for other EP-PCR studies [44]; all transitions and transversions are represented, though T>G and G>C are the rarest and G>A are the most frequent (S1 Fig). The frequency of variant detection outside the mutagenized region, but within the sequencing amplicon was substantially lower (0.097 per 1000 nt, SD = 0.108) and followed no significant nucleotide preference. The sequenced library included all but two of the 4,887 possible SNVs in the 543 bp *CHEK2* ORF.

To screen for *CHEK2* variants that retain function we selected yeast transformants that were able to grow in the absence of *RAD53* and determined the frequencies of the remaining SNVs by NGS (Methods). For each individual SNV, we compared the average frequency before selection (*i.e.*, in the library) to the average frequency after selection and calculated a *rad53* complementation score (RCS): log2 of the ratio of frequencies after/before selection. Lower ratios or more negative RCSs were interpreted to indicate that the SNVs were depleted after selection as expected for variants that are detrimental to CHK2 function. Globally, 4,885 of the 4,887 possible SNVs in the *CHEK2* ORF could be compared in this way; one SNV was missing from the library, and another was not detected in any of the four independent replicate screens (S1 Table). The SNVs that were evaluated represented all but one of the 3,913 possible unique coding outcomes, including 195 stop codons, 530 synonymous changes, and 3,187 non-synonymous changes (S2 Table). The average RCS for all variants was -0.94 (range -7.30–5.20).

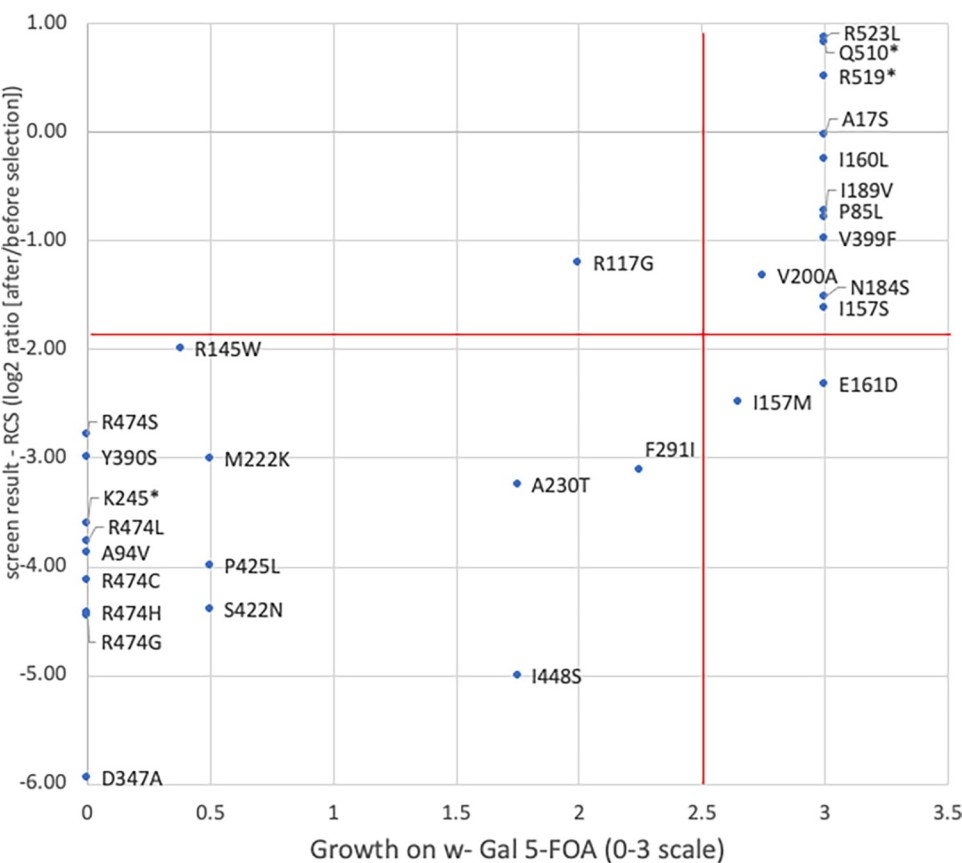

**Fig 1. Screen results for individually tested *CHEK2* variants.** Yeast *rad53* cells containing the *pURA3::RAD53* plasmid and expressing the indicated individual *CHEK2* variants were plated on media containing 5-FOA to select for loss of *pURA3::RAD53*. Growth is plotted in the X axis on a 0–3 scale where 0 is no growth (equivalent to vector only) and 3 is maximum growth (equivalent to wild-type *CHEK2* growth). Growth of 2.5 or less (vertical red line) was considered defective. Screen depletion values of each variant are plotted on the Y axis. Subsequent analyses indicate that most variants that are known to be functionally defective had depletion scores below -1.9 (horizonal red line).

To begin to test the assertion that loss of *CHEK2* function correlates with screen depletion, we constructed 30 variants and tested them individually for *rad53* complementation. We found that the 9 variants that did not fully complement *rad53* in individual assays were more depleted in the screen (ave. RCS -3.64, SD = 1.12) than were the 11 variants that fully complemented *rad53* in individual assays (ave. RCS -0.76, SD = 1.10) (Fig 1). These results suggest that depletion in the screen can be used as a predictor of variant function.

## Correlation of rad53 Complementation Screen Results with CHK2 Function

To see how well the screen results correlate with previous annotations of pathogenicity, we looked at SNVs in ClinVar [35]. ClinVar currently has information for 1,418 *CHEK2* SNVs, including 98 that are annotated as being pathogenic or likely pathogenic, 1,268 that are variants of uncertain significance (VUS), and 58 noted as having conflicting information regarding pathogenicity; ClinVar has no information on 2,237 of the non-synonymous SNVs that we evaluated in our screen. Consistent with the predictive power of our screen, the pathogenic variants were significantly more depleted (mean RCS -3.15, stdev = 1.49) than the VUS (mean

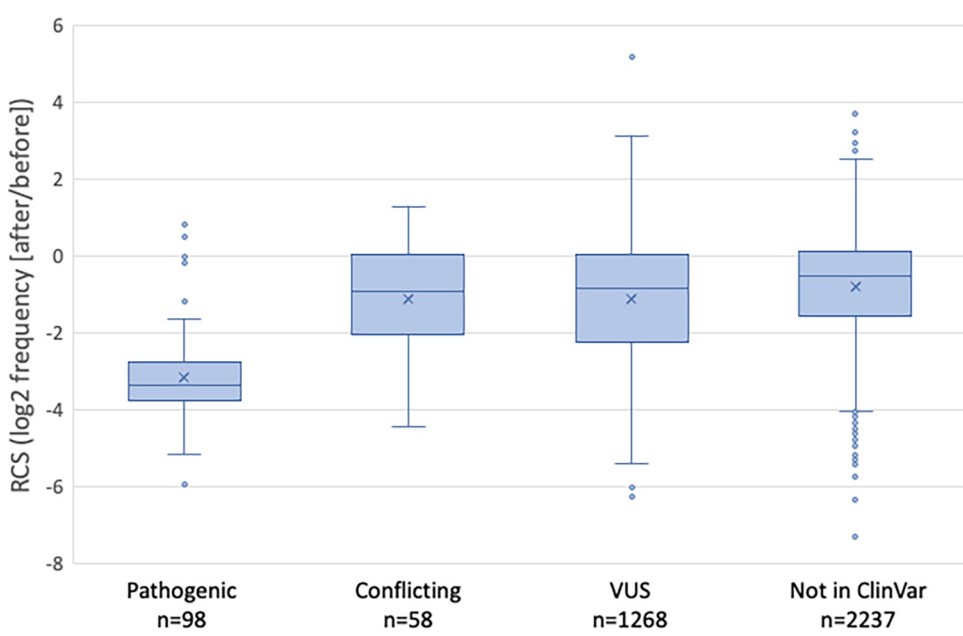

**Fig 2. *rad53* complementation screen scores (RCS) correlate with ClinVar annotations and expected variant effects.** Boxplots show the RCS for SNVs annotated in ClinVar as pathogenic or likely pathogenic (Pathogenic), with conflicting information about pathogenicity (Conflicting), and variants of uncertain significance (VUS). Also shown are the screen results for 2237 non-synonymous variants that are not listed in ClinVar. The differences between the means (marked with an X) of the pathogenic group and any other group are significant (P values $<10^{-14}$).

RCS -1.10, stdev = 1.54) or the variants with conflicting annotations (Fig 2). The VUSs as well as the SNVs that are not in ClinVar had a wider distribution of depletion values in the screen, consistent with the likelihood that only a fraction of these may be functionally defective.

Most (92/98) of the *CHEK2* SNVs that are considered pathogenic or likely pathogenic in ClinVar create premature nonsense (stop) codons in the ORF. To further validate our screen, we compared the results for SNVs that generated all 195 stop codons to SNVs that result in synonymous or non-synonymous codon changes (Fig 3). As expected, stop codons variants were significantly depleted in the *rad53* complementation screen, while the synonymous variants were not. Curiously, this analysis also showed that several stop codon variants were outliers in the screen, with relatively modest levels of depletion. To take a closer look at these, we plotted the depletion scores for all stop codons along the primary axis of the 543 residue CHK2 protein (Fig 4). This revealed that depleted stop codons were evenly distributed through most of the ORF, yet most of the outliers resided beyond codon 500. The average depletion score for the stops in the first 500 codons (-3.26, stdev = 0.91) was significantly different from that of the last 43 codons (0.42, stdev = 0.84; p $3.2x10^{-23}$), which mimicked the synonymous SNVs (Fig 3). Using individual clones, we confirmed that variants (p.Gln510stop) and (p. Arg519stop) were indistinguishable from wild-type in *rad53* complementation (Fig 1). Combined, these results strongly suggest that premature termination beyond residue 500 does not adversely affect the ability of *CHEK2* to complement *rad53* (discussed further below).

## Classification of Non-Synonymous SNVs as functionally Damaging or Tolerated

Having established that the results from the *rad53* complementation screen correlate well with known function or pathogenicity, we used the screen data to functionally classify the 3,187

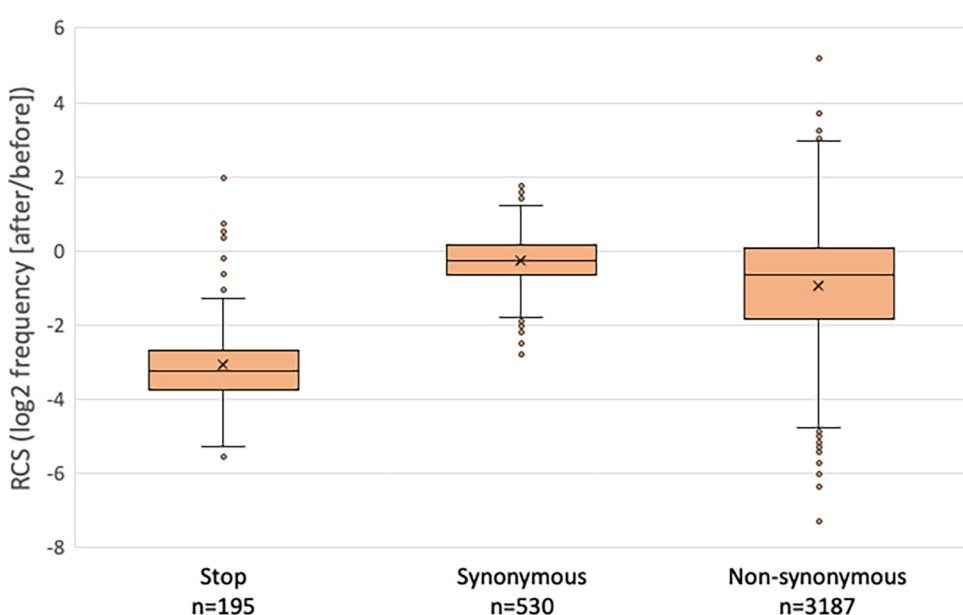

**Fig 3. *rad53* complementation screen scores (RCS) correlate with expected variant effects.** RCS for variants that result in the 195 possible stop codons, 530 synonymous, and 3188 non-synonymous changes. For this analysis, the SNV frequencies were combined for all SNVs that led to the same protein outcome (stop, synonymous, and each non-synonymous amino acid change). The differences between the means (marked with an X) of the stop codon group and any other group are significant (P values $<10^{-80}$).

unique amino acids changes that result from non-synonymous SNVs. Here we refer to variants that appear to reduce CHK2 function in the yeast assay as 'damaging' and those that are functionally indistinguishable from wild type as 'tolerated'. We tested various machine learning algorithms optimized for accuracy using as training data the 181 stop variants in the first 500 codons, presumed to be damaging, and the 587 synonymous variants also in the first 500 codons, presumed to be tolerated (see Methods). We found that the only data feature that was useful in predicting function was the RCS depletion score (log2 frequency [after/before selection]). Moreover, a logistic regression model with an RCS cutoff of near -1.9 achieved the optimal accuracy, correctly classifying 174/181 stop codons as functionally damaging and 480/487 synonymous variants as tolerated, for an overall prediction accuracy of 97.9% (654/668) (S2 Fig). For the synonymous variants beyond residue 500, which were not used in the training data, the classifier correctly labeled 43/43 as tolerated. In all, we functionally classified 3,579 non-synonymous SNVs resulting in 3,187 unique amino acid changes, including 770 amino acid changes labeled as damaging and 2,417 amino acid changes labeled as tolerated. These results are summarized in an expandable variant effect map (S3 Fig) and are deposited into MAVdb [45] (accession urn:mavedb:00001203).

Several functional studies of CHEK2 variants have included an "intermediate" category purported to represent variant function that is intermediate between wild-type (tolerated) and complete loss of function (damaging). To see if our *rad53* complementation score could be used to predict intermediate function, we looked at 443 variants that have been tested in another functional assay that includes an intermediate category. There was no correlation between the RCS and the intermediate category of the other functional assays (S4 Fig), demonstrating that the variants classified as intermediate by other assays could not serve as gold standard training data to classify intermediate function based on the RCS. It is worth noting that the intermediate category appears to be the least reliable based the lack of agreement among

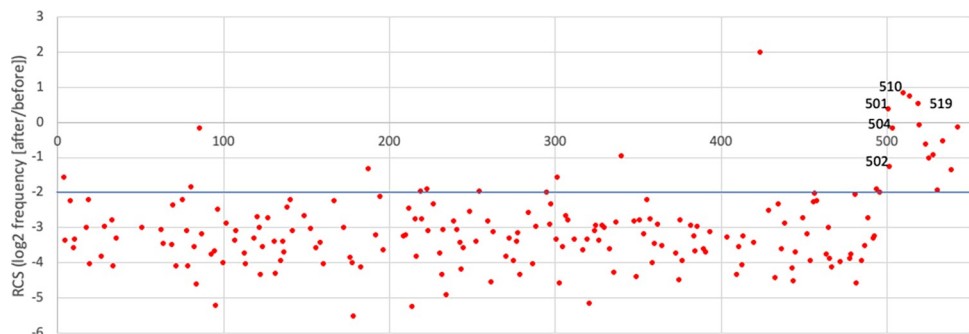

**Fig 4.** Stop variants. Screen scores for the 195 possible stop codons plotted along the CHK2 protein primary axis from residue 1 (left) to residue 543 (right). All of the stop codons beyond residue 500 have screen scores >-2.0 (line highlighted for emphasis).

different functional studies (discussed in detail below). For example, of the 108 variants scored as intermediate in a least one of the other 14 functional assays and with at least one other assay to compare with (not including our assay), 94% (101/108) of the intermediate calls disagree with another assay.

## Functional consequences of missense variants

The positions of putative damaging and tolerated variants predicted from the screen correlate well with previous knowledge about the structure and function of CHK2. We looked at the frequency of damaging and tolerated variants at each position across the length of the protein, or the average RCS score of all variants at each position, and several patterns emerged (Figs 5, and S5). The number of different amino acid changes tested at each residue varies from 4 to 7, distributed uniformly across the protein. While damaging variants are found throughout the protein, they are rarest at the N–and C-terminal ends. Residues 1–69 and 506–543 are thought to be disordered regions, both by prediction algorithms [46] and by structural analyses [47]. The fraction of protein variants that are damaging in these two disordered regions is 7.2% (29/405) and 10.6% (23/217), respectively, while the fraction of variants that are damaging across the entire protein is 24.2% (770/3187). The average RSC for all variants in these two disordered regions also shows that most variants are tolerated (Figs 5C and S5). The amino terminal disordered region also includes the SQ/TQ cluster domain (SCD) from residue 19 to 69, which has several serines (S) or threonines (T), and seven instances of S or T followed by a glutamine (Q). This region includes several SQ or TQ residues that are phosphorylated in response to DNA damage, including the main ATM target, T68 [2]. Phosphorylation of T68 promotes dimerization interactions between the SCD of one molecule and the FHA domain of another molecule, which in turn leads to autophosphorylation and activation of kinase activity [2–4]. While the SQ and TQ motifs are highly conserved, T68 specifically is not conserved in the CHK2 orthologs in *S. cerevisiae* (Rad53), *C. elegans*, *Drosophila*, or *S. pombe*, though these proteins do have other SQ and TQ in the region. In our screen, all 5 amino acid changes at residue T68 were tolerated, consistent with the previous finding that T68A was tolerated in another yeast assay [34]. Moreover, all variants in the other 6 TQ/SQ residues were also individually tolerated in our screen. Combined, these results suggest that phosphorylation of T68 or any other single TQ or SQ residue in the SCD of CHK2 is not required for *rad53* complementation in yeast. This is consistent with studies showing that the Rad53 SQ/TQ residues are not required to promote growth, while any one of the first four SQ residues in Rad53 is sufficient for DNA-damage-induced activation [48].

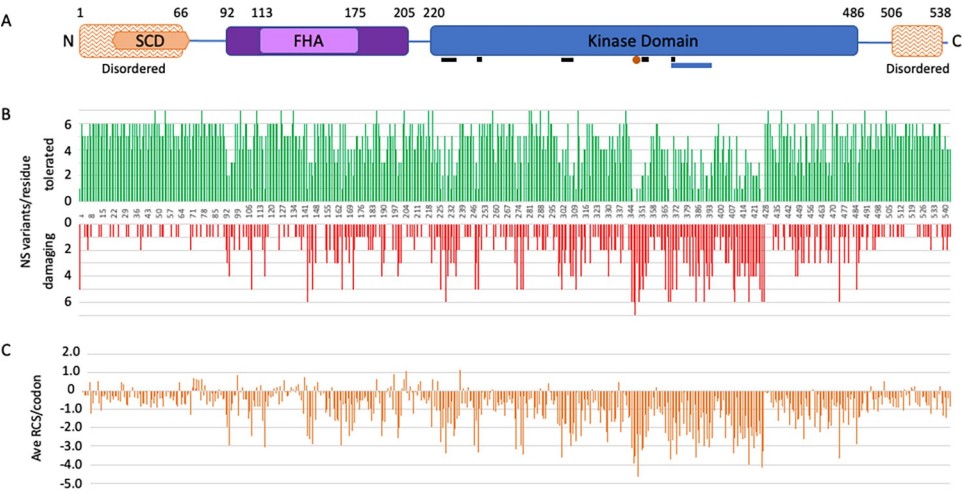

**Fig 5. (A)** Primary domain structure of the 543-residue CHK2 protein. Residues 1–66 and 506–538 are predicted to be intrinsically disordered regions [46]. The N-terminal disordered region overlaps with the SQ/TQ Cluster Domain (SCD) from residue 19 to 69, which contains seven SQ or TQ sites [2]. The forkhead-associated (FHA) domain extends from residue 113 to 175, according to sequence alignments [62]; a larger region from 92 to 205 has been characterized structurally as the FHA domain [47,63]. The conserved kinase domain extends from residue 220–486 [49,50]. Also shown are conserved ATP binding pocket residues (black bars under kinase domain), the proton acceptor residue D347 (orange dot), and the activation loop from residue 368–394 (blue bar under kinase domain). **(B)** Frequency of damaging or tolerated protein variants at each residue. The numbers of non-synonymous (NS) variants at each residue are plotted along the CHK2 primary axis as in (A) for tolerated variants (upper) and damaging variants (lower). This reveals individual residues and regions where variants appear to be infrequently tolerated. **(C)** The average RCS score for all variants at each codon is plotted along the CHK2 primary axis as in (A).

The first region with a high frequency of damaging variants was the FHA domain (aa92-205), with 21.5% (146/672) of the variants damaging; 51% (34/66) of the variants in FHA residues that are 100% conserved are damaging. Not surprisingly, the kinase domain (residues 220–486) has the highest fraction of protein variants that are damaging, at 34.6% (559/1562) (Figs 5 and 6, and S5). Furthermore, the frequency of damaging variants within the kinase domain is higher in amino acids and clusters that have conserved functions (Fig 5). For example, in the conserved ATP-binding pocket (Figs 5 and 6) [49], 52% (44/84) of the variants are damaging. Among the variants in the 46 kinase domain residues that are invariant from yeast to human, 69.1% (193/265) are damaging. No variants were tolerated in the proton acceptor residue (D347), while only 2 variants were tolerated in the two other conserved catalytic residues, K249 and D368 [50]; the two tolerated variants, K249R and D368H, have marginally low RCS of -1.55 and -1.78, respectively. In the rest of the activation loop or T loop (residues 368–394), most of the variants are damaging, and no variants were tolerated in residues F369, G370, C385, Y390. Four out of five variants were damaging in each of the autophosphorylation sites, T383 and T387. In the two residues that contact pT383 and pT387 in the dimer, R346 and K373, only one variant was found to be tolerated, the conservative variant K373R. A second region within the kinase domain, from V408 to F427, had a preponderance (70%) of damaging variants and low RCS (ave. -2.47), visible in Fig 5.

## The damaging classification correlated well with previous annotations

Previous functional assays for 162 *CHEK2* missense variants generated by SNVs were summarized in [51] (S3 Table). These assays include complementation of yeast *rad53* mutants either in the presence or absence of DNA damaging agents [24,32–34], phosphorylation of various CHK2 substrates *in vitro* or in mammalian cells [23,26,29–31,52,53], DNA damage responses

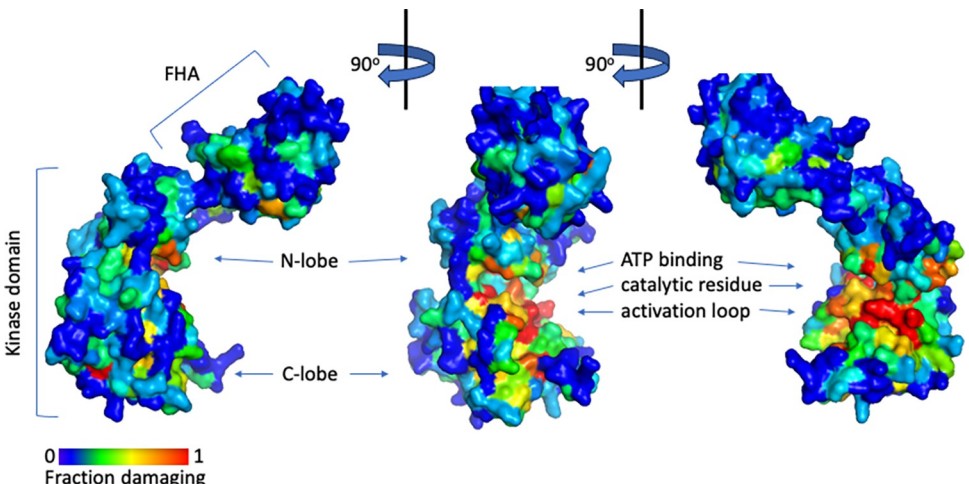

**Fig 6. Space-filled peptide backbone structure of CHK2 from three angles with backbone residues colored according to the fraction of variants at that residue that are damaging.** The Protein Database structure (3I6U) is from residues 89 to 501, including the FHA domain and the kinase domain with its conserved N-lobe and C-lobe forming an active site cleft between them; the structure lacks the unstructured N–and C–termini [47]. The active site region containing the ATP binding pocket, catalytic residue, and activation loop (partially shown) contains many residues unable to tolerate variants (yellow to red).

in cells [53,54], and protein stability [30,31]. The results of these various assays have been used to functionally classify each variant as either damaging to the protein's function, tolerated (sometimes referred to as "functional"), or with some intermediate level of function that is less than that of wild-type. The various assays do not always agree on the results for any given variant. The disagreements may be due in part to the fact that the different assays are probing different aspects of CHK2 function; in some cases, however, disagreements between assays may be due to varying levels of miss-classification in each approach. Nevertheless, our screen classification agrees well with the other functional studies. For example, of the 53 previously tested variants that we classified as damaging, only 5 were classified as tolerated by another study, and in all cases it was just one other study (see below). None of our damaging calls were contradicted by more than one study.

Roeb *et al.* [33] and Delimitsou *et al.* [34], assayed *rad53* complementation in the presence of DNA damaging agents to test 26 and 119 *CHEK2* missense SNVs, respectively, and functionally classified each as either damaging, tolerated, or intermediate. While these two assays were very similar to each other, the results agreed on only 7 of the 17 variants that were tested in common. In 6 of the 10 differences, one or the other study classified the variant as intermediate, a classification that we did not attempt to make and that may be more prone to error. Our data agree with Delimitsou *et al.* [34], for 89/119 (75%) of the variants; half of the disagreements were classified as intermediate by Delimitsou *et al.*[34]. In contrast, our data agree with Roeb *et al.* [33], for only 9/26 (35%) of the variants, and again nearly half of the disagreements were classified as intermediate by Roeb *et al.* [33]. There were only three variants where our classification (tolerated) was different from that of both other studies (damaging): R117G, G167R, and Y424H. Interestingly, when we tested R117G individually, it supported less than wild-type growth even though that difference was not detected in our large-scale screen (Fig 1).

Ten additional studies summarized by Boonen *et al.* [51] evaluated a total of 75 SNV variants in functional assays. Our functional classification agreed with 75% of the 132 individual functional assays; again, over half of the differences included assays that found an intermediate

phenotype. For the 71 variants that we found to be damaging, all but one other assay that was performed also found to be damaging, and the one exception found the variant to be intermediate. Boonen et al. [51] suggested a counting system to settle differences between assays. At that time, 48 variants had been tested in more than one assay. Among these 48, our damaging calls agree with the consensus for all but one variant where the consensus was intermediate. Our tolerated calls agree with most of the consensus calls and half of the disagreements are with a consensus call of intermediate. I157T is a well-studied example of a variant that we classified as tolerated but where there is disagreement among assays: eight other assays also called I157T tolerated, while one [26] called it intermediate and one [33] called it damaging. Only seven of our calls, all tolerated, differed from the consensus of multiple assays. For two of those, adding our tolerated call to the calculation of a consensus would change the consensus to tolerated. Thus, other assays provide strong evidence that only 4 of our calls are wrong: our tolerated designation of R117G, D162G, G167R, and S428F is countered by multiple other assays that found them to be damaging.

Another recent large study by Stolarova *et al.* [22], functionally characterized 427 CHEK2 VUS selected by the international ENIGMA (Evidence-based Network for the Interpretation of Germline Mutant Alleles) consortium for variants with an association to an increased risk of breast cancer. That study used two parallel functional assays that quantified the phosphorylation of the KRAB-associated protein (KAP1) at S473 and the autophosphorylation of CHK2 at S516 to characterize 427 missense SNVs as functionally wild-type, intermediate, or impaired. The two assays were concordant for 79.1% of the variants. When compared to our results, the phosphorylation of KAP1 and autophosphorylation of CHK2 experiments agreed in 322/427 (75.4%) and 318/427 (74.5%) of the variant classifications, respectively. This analysis does not include those classified as functionally intermediate, for which there is substantial disagreement even within the Stolarova *et al.* [22] assays. For example, while the KAP1 and CHEK2 phosphorylation assays agreed on 6 intermediate calls, they disagreed on the remaining 71 variants where one or the other assay called intermediate function: these account for 79% (71/90) of the disagreements between those two assays. Compared with the concordant KAP1 and CHK2 phosphorylation data, our functional classifications were consistent in 337/427 (78.9%) of the variants.

## Functional assays and variant effect predictors

Several algorithms have been developed to predict the function or pathogenicity of protein variants in the absence of functional data. We compared our classifications to three variant effect predictors (VEPs) that either combine various algorithms (REVEL) [55] or various structural, functional, or conservation data (HELIX [23] and AlphaMissense [56]) to predict altered protein function and therefore pathogenicity. Each of these VEPs score variants on a scale of 0 to 1 reflecting the probability of the variant being pathogenic or damaging to protein function. The REVEL, HELIX and AlphaMissense scores correlated with the *rad53* complementation scores from our screen (Fig S6 and S3 Table). The average REVEL, HELIX, and AlphaMissense scores for variants we found to be damaging were 0.566, 0.659, and 0.73, respectively, while the average scores for the tolerated variants were 0.304, 0.309, and 0.36, respectively. REVEL correctly predicted 461/770 (59%) of the functionally damaging variants while HELIX correctly predicted 532/770 (69%). These results underscore the importance of experimental verification of function predictions and the variability between these prediction algorithms.

While it remains to be determined whether computational predictions like AlphaMissense are more accurate at assigning function to variants than are experimental functional assays [57], the predicted classifications can still be used as a benchmark to compare the various

functional assays with each other. As we observed when comparing between functional assays, the ability of AlphaMissense to accurately predict the 'intermediate' phenotype was very poor (often limited to 11–15% accuracy). Thus, we only compared the ability of AlphaMissense to predict damaging or tolerated variants in the functional assays and did not consider variants that were classified as intermediate by AlphaMissense or experimentally. AlphaMissense was able to accurately predict 71.1% of the variants that we found to be damaging or tolerated. Similarly, AlphaMissense accurately predicted from 71–81% of damaging or tolerated variants in other, smaller datasets (S7 Fig and S3 Table). Interestingly, AlphaMissense appears to be particularly accurate at predicting damaging variants. For example, AlphaMissense accurately predicted all 82 variants that were classified as damaging by more than one functional assay. In contrast, AlphaMissense accurately predicted only 73.5% of the 174 variants that were classified as tolerated in more than one functional assay. These results also support the use of multiple functional assays to accurately classify the damaging variants.

## Discussion

The fact that several known loss-of-function CHK2 variants are associated with increased cancer risk leads to the expectation that any CHK2 loss-of function variant would also contribute to cancer risk. This forms the rationale for determining the functional consequences of CHK2 variants, either experimentally with functional assays or computationally with VEPs. Applying functional data, therefore, can contribute as evidence to the clinical classification system put forth by the ACMG, even for rare variants that have no segregation data [28]. The high throughput approach presented here provided a functional assay for most of the CHK2 variants that can be generated by SNVs. The screen evaluated the functional impact of 4,885 of the 4,887 possible SNVs in the *CHEK2* ORF, which included all but one of the 3,913 possible unique protein-coding changes. This included functional data for the 1,268 VUS in ClinVar along with an additional 2,237 non-synonymous variants. The data also provide additional supporting evidence for the 58 variants that have a ClinVar clinical classification as "conflicting interpretations of pathogenicity", and for 443 missense variants that have been tested in at least one other functional assay.

A problem that has emerged in using computational or experimental data to determine variant function is that the various assays or prediction methods do not always agree on the classification. There are several possible explanations for these disagreements. For some variants, the explanation lies in how different studies report variants and whether the classification systems are based on two (tolerated or damaging) or three (including intermediate) functional classes. For example, of the five variants annotated in ClinVar as "benign/likely benign", our screen classified one (I448S) as functionally damaging. In contrast, four other functional assays all found this variant to be intermediate (meaning somewhat damaging), while the VEP algorithms REVEL, HELIX, and AlphaMissense gave this SNV a score of 0.142 (tolerated), 0.348 (tolerated), and 0.546 (ambiguous), respectively. Interestingly, while our screen was incapable of discriminating damaging from intermediate function, a test of the I448S individual clone did show intermediate levels of *rad53* growth complementation (Fig 1). Another example is I157T, the variant most studied across CHK2 functional screens (12 total assays; S3 Table). Our screen as well as six others found this variant to be tolerated; however, three assays functionally classified it as intermediate, and one classified it as damaging. In agreement with the majority of the functional screens regarding I157T, the three VEP scores (0.316, 0.167, and 0.229 all result in a tolerated classification. I448S and I157T may be examples of variants that have a truly intermediate effect on function, which different assays interpret differently based on their different sensitivities and classification systems.

Disagreements among individual functional screens may be due to the differences in the model organism or in the assays themselves. Different functional assays may assess distinct aspects of CHK2 function. For example, our assay measured the ability of CHK2 variants to promote growth in the absence of Rad53 and DNA damaging agents, while other assays in yeast measured the ability of CHK2 variants to overcome DNA damage in a strain that does not require *Rad53* or CHK2 for growth. Thus, our screen may miss damaging variants in positions in CHK2 that are not required for *rad53* growth complementation, but instead may be required for activity in response to DNA damaging agents. It is also possible that differences may arise for variants that affect CHK2 activity toward one subset of its targets and not another. Furthermore, while *Saccharomyces cerevisiae* has been used to successfully assess human genetic variants it does have its limitations. Some ORF variants may impact transcript splicing, protein binding sites, thermostability, or subcellular localization differently in yeast cells compared to mammalian cells and *in situ*. For example, our results suggest that beyond residue 500, premature termination does not negatively impact *CHEK2's* ability to complement *rad53*. However, residues 515–523 contain a nuclear localization signal (NLS) that is required for nuclear localization in human cells [58], and this was confirmed by the ENIGMA study showing that truncations at position 519 (p.R519X) or 523 (p. R523fsX), or NLS missense variants p.Arg521Gln and p.Arg521Trp, did not localize to the nucleus in human cells. p.Arg521Trp was tolerated in our screen and in another yeast assay [34]. Combined, these results suggest that the C-terminal NLS is required for nuclear localization in human cells, but not in yeast cells. Yeast cells may be capable of using one of the other two consensus NLS in CHEK2, even though they are not required in human cells [58]. Given the lack of requirement for the C-terminal 43 amino acids for function in our yeast assay, it is puzzling that 25 of the 249 missense variants in that region were found to be damaging, well beyond the expected false positive rate in our assay. An explanation for this apparent discrepancy may require additional functional assays for these variants in both human cells and yeast.

A common solution to the problem of discordance between functional assays is to seek consensus, as suggested by Boonen *et al.* for CHK2 functional assays [51]. In this approach, functional classifications confirmed by more than one assay or VEP are taken to be higher quality. Such an approach is supported by the finding that concordant data from multiple assays has significantly more overlap with independent VEP predictions than data from a single assay. The value of a second, confirming assay is further supported by an analysis of ENIGMA data and two functional assays (KAP1 phosphorylation and CHK2 phosphorylation) carried out by Stolarova *et al.* [22], which shows that CHK2 variants confirmed to be damaging in more assays are associated with increasing risk for breast cancer. For example, the 140 variants that either of their functional assays classified as damaging, even if the two assays were discordant, showed a combined increased risk for breast cancer (OR 2.29; 95% CI 1.97–2.66), while the 91-variant subset of these that were confirmed to be damaging in our screen showed a marginally increased risk (OR 2.95; 95% CI 2.36–3.69), and the risk associated with 74 variants confirmed to be damaging in all three assays was even higher (OR 3.05; 2.39–3.90) (S3 Table and [22]). These results support the continued application of functional assays to establish a high confidence clinical classification of variant effects. Our study provides an additional piece of functional data that may be combined with other evidence for the classification of *CHEK2* variants.

## Material and methods

### Plasmids

pRS314-GU-HR is a *TRP1*, CEN/ARS yeast plasmid with the *GAL1* promoter driving expression of inserts. pRS314-GU-HR was made from pRS314-GU (a gift from Phil Heiter), a derivative of pRS314 [59] in which the *GAL1* promoter was inserted at one end of the multiple

cloning site and the *URA3* promoter at the other end in the opposite orientation to serve as a terminator. The oligonucleotides 314GU-A (CGATTTGACTGTATCGCCGGAATTCGGGGC CCGGATCCCTGCAGCCAAGCTAATTCCGGT) and 314GU-B (GATCACCGGAATTAGC TTGGCTGCAGGGATCCGGGCCCGAATTCCGGCGATACAGTCAAAT), containing 5RT and 3RT flanking and EcoRI and BamHI cloning sites were annealed and ligated into the ClaI-BamHI interval of pRS314-GU to generate pRS314-GU-HR. The *CHEK2* open reading frame corresponding to transcript NM_007194, encoding *CHEK2* isoform a, within a Gateway entry clone was changed by site-directed mutagenesis to eliminate the BamHI with a silent single-nucleotide change (ORF T1410->C), to make pENTR-CHEK2-BamHI. pENTR-CHEK2-BamHI was then used as a PCR template with primers that added an EcoRI site immediately adjacent to the ATG and a stop codon and BamHI site immediately after the last codon. The resulting EcoRI-BamHI fragment was cloned into EcoRI/BamHI cut pRS314-GU-HR to make pRS314-GU-HR-CHEK2-BamHI. Site-directed mutagenesis was carried out on pENTRY-CHEK2-BamHI to create the 30 variants shown in Fig 1. We subsequently learned that the absence of a stop codon in the original pENTRY-CHEK2 diminished the ability of *CHEK2* to complement yeast *RAD53* mutants when cloned into a yeast vector. Thus, we converted all of the pENTRY-CHEK2-BamHI variants to versions with the natural stop codon of *CHEK2* as follows: We PCR amplified wild type *CHEK2* and each variant from pENTRY-CHEK2-BamHI with primers that added an attB1 site to the 5' end and a stop codon and attB2 to the 3' end, then used the PCR product in a BP reaction into pDONR221. The resulting pENTRY-CHEK2-stop-BamHI variants were then cloned into the yeast vector pAG414GAL (a gift from Susan Lindquist; Addgene plasmid # 14143) using the LR reaction. All *CHEK2* clones were verified by sequencing.

## RAD53 Complementation assays

For complementation of the *rad53* deletion, we used yeast strain YGB272 (*MAT**a** rad53::HIS3 pRAD53::URA3*), which was derived from a *MAT**a**/MATα RAD53/rad53::HIS3 SML1/sml1-1* heterozygote (W303 background) transformed with pJA92 encoding *RAD53* and *URA3* [60] (diploid and plasmid kindly provided by Dr. Stephen Elledge). To test individual *CHEK2* variants for *rad53* complementation, YGB272 was transformed with pRS314-GU-HR or pAG414-GAL containing wild-type *CHEK2* or variants and transformants were selected on synthetic complete media lacking tryptophan (-trp) and containing 2% glucose (Glu). YGB272 requires *RAD53* for growth and will not grow on 5-fluoroorotic acid (5-FOA), which is toxic to strains containing *URA3* [61]. Previous studies have shown that expression of wild-type human *CHEK2* in such a strain will allow it to lose the p*RAD53*::*URA3* plasmid and to grow on 5-FOA [1]. Individual transformants were picked and grown in 2 ml of -trp liquid media supplemented with 2% galactose and 1% raffinose (Gal/Raf) at 30 degrees with shaking to saturation. Cells were then serially diluted in sterile water and 5 μL of each dilution was plated to -trp Gal/Raf plates supplemented with 1mg/ml 5-FOA (Gold Biotechnology). After 4 days at 30 degrees, the spots were scored for growth as follows: 0 for no growth, 1 for very weak growth, 2 for moderate growth, and 3 for complete growth. Strains with the vector only always scored 0 while strains expressing wild-type CHK2 always scored 3.

## *CHEK2* Variant Library

The library was constructed by first PCR amplifying the *CHEK2* ORF from pENTR-CHEK2-BamHI with primers (HsCHEK2_5RT and HsCHEK2stop_3RT) that added an EcoRI site immediately adjacent to the ATG and a stop codon and BamHI site immediately after the last codon, respectively, using high-fidelity PCR conditions (Platinum Super-Fi polymerase,

Invitrogen). The fragment was gel purified and used as template in error-prone PCR (EP-PCR) using the GeneMorph II kit and Mutazyme II polymerase (Agilent Technologies) and the primers (5RT 3RT). Ideal EP-PCR conditions were established in pilot assays with *CHEK2* and other ORFs in both complementation assays and yeast two-hybrid assays and by sequencing to maximize the number of clones with a mutation while minimizing the number of mutations per clone. The final EP-PCR conditions used to make the *CHEK2* library were: five separate 50 μL reactions with 0.8 mM dNTP mix, 0.25 μM each primer, 100 ng template DNA, and 0.017 units/μL of Mutazyme II and these were treated for 27 cycles of 95 degrees C for 30 seconds, 57 degrees for 30 seconds, and 72 degrees for 2 minutes. The EP-PCR products were purified through a GeneJET PCR purification column (ThermoFisher), cut with hi-fidelity EcoRI and BamHI (New England Biolabs), and gel purified. The fragments were then ligated into pRS314-GU-HR linearized with EcoRI and BamHI. Bacterial strain OmniMax (ThermoFisher) was electroporated with the ligations and plated onto 22cmx22cm LB amp plates. Plasmid DNA was prepared from 22,400 transformants.

## Library screening

The pRS314-GU-HR-CHEK2 EP-PCR library DNA was used to transform strain YGB272 and 62,000 trp+ colonies were obtained on -trp Glu plates. The colonies were pooled and frozen in aliquots containing 2x10e7 colony forming units (cfu)/100 μL (after the freeze/thaw cycle). Two aliquots (LibA and LibB) were used to prepare DNA for sequencing. Complementation screens were carried out in quadruplicate (screen S1 to S4), each by inoculating 100 ml -trp Gal/Raf with 100 μL of a library aliquot and growing at 30 degrees with shaking for 24 hours to induce expression of the *CHEK2* ORF; based on OD600 this resulted in a 21-fold amplification of cells. 4x10e7 cfu were plated on -trp Gal/Raf supplemented with 1mg/ml 5-fluoroorotic acid (5-FOA) (Gold Biotechnology) and were grown at 30 degrees for 96 hours. In the quadruplicate screens, 28,000, 26,600, 26,600, and 30,400, 5-FOA-resistant colonies, respectively, were collected and frozen aliquots were made containing 5x10e6 cfu/100 μL in -trp Gal/Raf (after the freeze/thaw cycle). Aliquots of the unselected library transformants (LibA and LibB) and the four 5-FOA-resistant yeast (S1 to S4) were thawed and used to prepare crude plasmid DNA using zymolyase and Qiagen miniprep columns. To create amplicons for next generation sequencing, the mutagenized inserts were amplified with vector primer sequences from 46 bp upstream of the *CHEK2* ATG and 144 bp downstream of the *CHEK2* stop codon.

Sequencing libraries were constructed with the Nextera XT DNA Library Prep kit with Nextera XT Indexes (Illumina), pooled, and sequenced on a NextSeq 500 to obtain 76 bp paired-end reads. Sequence data was processed within Geneious using BBDuk to trim adapters, filter out quality scores below 30 and sequences less than 60 bp, then the Geneious algorithm was used to map reads to the *CHEK2* amplicon and to detect variants. The number of mapped reads for each library ranged from 2.5 to 5.6 million. Subsequent analyses focused on the single nucleotide variants (SNVs). All SNVs in the *CHEK2* ORF were represented in the library sequences, except one (CDS position 1129 A->C, p.Ile374Leu). For each SNV and each library we calculated the frequency as the number of reads with that SNV divided by the coverage at that nucleotide position. For every SNV present in the library, we calculated the average of frequencies in the two aliquots before selection (LibA and LibB) and the average of frequencies across the four screens, S1-S4 after selection. We then calculated a *RAD53* Complementation Score (RCS) as log2 (average frequency after/average frequency before selection). 669 codon changes (synonymous, non-synonymous, or stop) were represented by more than one (2–4) SNVs. For these we calculated the average frequencies and depletion scores across all SNVs leading to the same change.

## Supporting information

**S1 Fig. Nucleotide variant frequencies.** Variant frequencies are shown as the average of variant counts at each position divided by the sequencing coverage at that position x 1000. All frequencies were normalized to the most frequent wild-type nucleotide in the fragment being sequenced. Fragments include the 1629 bp mutagenized *CHEK2* ORF (ORF) and two regions of the sequenced amplicon that were not mutagenized, combining 46 bp upstream of the *CHEK2* ATG and 144 bp downstream of the *CHEK2* stop codon (no-EPCR). For each, the results are shown for the library after introduction into yeast (Library), and after selection for *rad53* complementation (Selected).
(TIFF)

**S2 Fig. Logistic regression model to classify non-synonymous variants as functionally damaging or tolerated.** (**A**) Training data, model parameters, and output applied to the *CHEK2* ORF. (**B**) Receiver operating characteristic (ROC) curve for the final model as applied to the training data. (**C**) Confusion matrix showing the numbers of training data (gained stops and synonymous SNVs) correctly classified (true positives, green) and incorrectly classified (false positives, red) by the model. Note that false positives in one class are equivalent to false negatives in the other class.
(TIFF)

**S3 Fig. A variant effect map summarizing the effect on function of all CHK2 protein variants from this study.** Residue positions and original amino acids are shown along the bottom while variants are listed at the left. Variants are colored according to their probability of being damaging (1.0 = red, 0.0 = blue), based on a logistic regression model of the screen depletion scores or RCS.
(TIFF)

**S4 Fig. Lack of correlation between the *rad53* complementation score and the intermediate functional classification of other functional assays.** In both graphs the *rad53* complementation score (RCS) score is on the X axis, and as a reminder a red vertical line shows the cutoff between damaging ($<$ -1.9) and tolerated ($>$ -1.9). On the y axis is the average score of other functional assays, where 1.0 is damaging, 0 is tolerated, and anything in between is intermediate (data in S3 Table). In both graphs, the variants with intermediate function in other assays are spread uniformly across the RCS scale. This is the case for the 443 variants that have been tested in at least one other functional assay (left graph) and for the subset of variants tested in the KAP1 kinase assay (by Stolarova et al. [22]).
(TIFF)

**S5 Fig. Frequencies of variants and damaging variants across the CHK2 protein.**
(TIFF)

**S6 Fig.** Comparison of functional classifications from this study to the prediction algorithms, REVEL [55] (A) and HELIX [23] (B), and Alphamissense[56] (C). In each, variants are scored on a scale of 0 to 1 (Y axis), with scores $>$ = 0.5 predicting pathogenic (damaging) and scores $<$0.5 predicting benign (tolerated). The average REVEL, HELIX, and Alphamissense scores (x in the boxplot) for variants we found to be damaging was 0.566, 0.659, and 0.73 respectively, while the average scores for the tolerated variants was 0.304, 0.309, and 0.36 respectively.
(TIFF)

**S7 Fig. Accuracy of AlphaMissense at predicting results of functional assays.** Overall accuracy: the fraction damaging, tolerated, and ambiguous or intermediate (if the assays included

such a designation) that were accurately predicted by AlphaMissense. The "Accuracy of damaging or tolerated only" ignores all variants designated as ambiguous or intermediate by AlphaMissense or the functional assay. The last row includes only variants tested in more than one assay, where all assays agree that they are tolerated or damaging.
(TIFF)

**S1 Table. *rad53* screen data for all *CHEK2* ORF SNVs**
(XLSX)

**S2 Table. Damaging and tolerated CHK2 protein variants.**
(XLSX)

**S3 Table. CHK2 variants and results from functional assays and VEPs.**
(XLSX)

## Acknowledgments

We thank Adnan Alazizi and the Genomics Services Center in the Center for Molecular Medicine and Genetics, Wayne State University School of Medicine for help with sequencing. We also thank Jeffery Tseng for adding our custom values to the CHK2 PDB file.

## Author Contributions

**Conceptualization:** George S. Brush, Michael A. Tainsky, Russell L. Finley, Jr.

**Formal analysis:** Claire E. McCarthy-Leo, Russell L. Finley, Jr.

**Investigation:** Claire E. McCarthy-Leo, George S. Brush, Russell L. Finley, Jr.

**Methodology:** Russell L. Finley, Jr.

**Resources:** Roger Pique-Regi, Francesca Luca.

**Supervision:** Michael A. Tainsky, Russell L. Finley, Jr.

**Writing – original draft:** Claire E. McCarthy-Leo, Russell L. Finley, Jr.

**Writing – review & editing:** George S. Brush, Francesca Luca, Michael A. Tainsky.

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
