## [Decision Letter · Decision Letter 0]

21 Feb 2024

Dear Dr Finley,

Thank you very much for submitting your Research Article entitled 'Comprehensive analysis of the functional impact of single nucleotide variants of human CHEK2' to PLOS Genetics.

The manuscript was fully evaluated at the editorial level and by independent peer reviewers. The reviewers appreciated the attention to an important problem, but raised some substantial concerns about the current manuscript. Based on the reviews, we will not be able to accept this version of the manuscript, but we would be willing to review a much-revised version. We cannot, of course, promise publication at that time.

If you decide to revise the manuscript for further consideration at PLOS Genetics, please aim to resubmit within the next 60 days, unless it will take extra time to address the concerns of the reviewers, in which case we would appreciate an expected resubmission date by email to plosgenetics@plos.org.

We are sorry that we cannot be more positive about your manuscript at this stage. Please do not hesitate to contact us if you have any concerns or questions.

Yours sincerely,

Haico van Attikum

Guest Editor

PLOS Genetics

David Kwiatkowski

Section Editor

PLOS Genetics

Thank you for submitting your manuscript "Comprehensive analysis of the functional impact of single nucleotide variants of human CHEK2" to PLOS Genetics. Reports from 4 reviewers have been received. and, on the basis of their comments, you are invited to submit a (major) revision of your work for further consideration at PLOS Genetics. Your revision should address all the points raised by the 4 reviewers (see their reports below).

Reviewer's Responses to Questions

**Comments to the Authors:**

Reviewer #1: The report by McCarthy-Leo et al is a substantial contribution to our understanding of the functional impact of SNV’s in CHEK2. The paper is detailed and pays considerable attention to considering the new data in the context of previous functional screens and application to clinical variant classification.

The authors present a model that categories variants, based on their RCS, as damaging or tolerated (which is supported by their data). However, comparison to previous reports that include a third intermediate category is difficult and this comparison is an important aspect of this report and its anticipated utility for variant classification. Have the authors tried to achieve a model with an intermediate category? If so, what was the outcome?

I have a small number of other minor comments and suggestions.

1. “Patient screening” might be better expressed as “susceptibility screening”

2. It could be made clearer that “patient samples” are blood samples not tumour samples.

3. “Cancer-causing” would be better expressed as “associated with cancer risk”

4. In many instances the term “mutation” could be replaced with pathogenic and likely pathogenic variants.

5. In results, first para, ; “all mutations” – are the authors referring to all possible nucleotide substitutions? …as the para goes on to describe that two SNVs were not identified in the sequenced library.

6. “We found that most of the individual variants that did not fully complement rad53 were depleted in the screen (ave. RCS -3.64, SD=1.12), while most of the individual variants that fully complemented RAD53 were not depleted…”. Please provide n for “most”.

7. Please clarify “The also data provide…”

8. ENIGMA does not themselves have patients - please edit.

9. “Conferred by” would be better expressed as “associated with”

Reviewer #2: In this manuscript, McCarthy-Leo et al. first reinvented the yeast RAD53 complementation assay, which they then used to evaluate all possible SNVs (-1) affecting the ORF in the human CHEK2 gene. The variants were generated using a well-balanced EP-PCR, and the PCR products were used to prepare constructs for yeast transformation and subsequent RAD53 complementation assay in the presence of 5-FOA. Next, they determined the abundance of individual SNVs by NGS before and after the complementation assay, which allowed them to discriminate between complementing SNVs (not depleted, retaining CHK2 kinase function) and non-complementing SNVs (depleted, indicating SNVs with impaired CHK2 function). Finally, the authors demonstrate the consistency of their results across the CHEK2 ORF and provide evidence for the correlation of their results with CHK2 function, ClinVar data, and data from previous analyses.

Overall, the data presented in this manuscript are of high quality and represent an original contribution, as no other AVE CHEK2 data have been published. Although the assay involves the yeast model and thus (as correctly discussed by the authors) pursues RAD53 complementation (which is not equivalent to the natural CHK2 function in human cells), this study will be of interest to many researchers from a broad spectrum of theoretical, preclinical, and clinical research. The work is technically robust and well suited for publication.

Minor points:

The authors provide some initial evidence of their assay using a set of 30 manually constructed variants (Fig. 1); however, 29 variants are shown in this figure.

Forkhead domain (FHA) should be described as forkhead-associated domain.

Regarding the comparison with the ENIGMA consortium study, the numbers in this section should be revised as 430 VUS were successfully analyzed by both functional assays in the ENIGMA study (not 427); moreover, the results in this manuscript are compared with 427 variants in the KAP1 assay, but with 327 variants in the CHK2 assay.

Authors should note in the Discussion section that the discordance between individual functional screens may also be caused by differences in the model organisms used for the functional assays (the current discussion focuses more on the comparison of different readouts in yeast studies). Whether this phenomenon could explain the discordant results between this manuscript and ENIGMA studies regarding NLS would be worth commenting, as all nonsense variants from S500 were found to have normal activity, in comparison to ENIGMA study, where all tested truncating variants up to (including) p.523fsX had impaired function due to aberrant localization (see Figure 3B in Stolarova et al., reference 26).

Finally, it should also be noted that for variants affecting pre-mRNA splicing, the results of functional synthetic assays may be very different from the biological reality in situ.

Reviewer #3: This study presents a deep mutational scan of nearly all possible single nucleotide substitutions in CHEK2 using a yeast-based complementation assay. These kinds of high-throughput experimental assays are becoming increasing important for variant analysis, with potential clinical implications. I do think the work is technically sound and potentially very useful to the community, and I believe it should be published in PLOS Genetics. However, there are a few issues that I think should first be addressed.

The paper positions itself well within the context of past work in the CHEK2 field, but there is little discussion of other work in the DMS/MAVE field. These high-throughput assays have become increasingly common in recent years, including several other yeast complementation DMS experiments, so it would be good to mention them and discuss how this compares. I would make sure to include the term DMS or MAVE in the abstract, to make it more discoverable.

I would love to see a variant effect map of amino acid substitutions (ie heat map with all missense variants colored by effect), which is typically included in most DMS studies. I find these to be more interpretable and useful than just showing the averaged variant effect per position.

This dataset is presented as being useful for screening VUS, but from the paper, it's difficult to assess how useful it actually is. Actually, the agreement with previous lower throughput assays doesn't seem great. Since really what people are interested in is performance on missense variants. It would be helpful to show a typic ROC AUC analysis on the known (likely) pathogenic vs (likely) benign missense variants from ClinVar. I know there aren't a lot, but it should still be performative, and then this could also be compared to the computational predictors. Note that even if this analysis shows that the experimental measurements are 'worse' than the computational methods, in my opinion that doesn't take away from the fact that this study is important and should be published because 1) DMS data represents independent information, 2) it is important for us to learn how well these types of assays perform using different genes and different experimental methodologies, and 3) many of the computational predictors are trained on human variants so their performance can be overstated (the exception being pure evolution-based methods).

Is it possible to provide any confidence/error estimates for the RCS scores? For example, I imagine library coverage is not perfectly uniform, so variants that have lower coverage in the library should have lower confidence. A lot of studies don't provide complete coverage due to such issues, so I can imagine some lower quality measurements were included to make the dataset almost complete. If you select a higher confidence, is the agreement with previous assays or ClinVar labels better?

The RCS scores should be deposited in MaveDB (www.mavedb.org) to improve accessibility.

Reviewer #4: The authors have provided a MAVE analysis of all CHEK2 single nucleotide variants using a high throughout growth assay in rad53 deficient yeast. The study is attractive because it goes well beyond all previous CHEK2 functional studies by evaluating all possible SNVs. The focus on growth rather than DNA repair is a useful addition to the compendium of assays now available for CHEK2 variants. However, the problem with this and other assays is that there is a high proportion of discrepant results. Perhaps as suggested by Boonen et al, and echoed by these authors, consistency in 2 or more high quality assays will be needed.

While interesting, there are a number of significant issues that must be addressed.

The biggest problem is that the analysis is limited to a binary cutpoint, which yields damaging and tolerant categories. The problem is that function is very clearly a continuous measure and studies of CHEK2 and other genes have shown that intermediate functional effects with associated intermediate risks of disease very clearly exist. The authors must recalibrate their assay and apply a two-component mixture model based on damaging and tolerant standards (perhaps nonsense and synonymous variants). The VarCall model that has been used for BRCA1 (Clark et al 2022) and BRCA2 could prove effective. Application of this model may help to resolve the high degree of discrepancy (25%) that the authors report for the KAP1 phosphorylation assay that is currently considered the standard for CHEK2 variant evaluation.

Supplementary Figure 2 should be a main figure in the manuscript but it must be improved to show the distribution of the stop codons, synonymous and missense variants. The current Figure 2 is insufficient for this purpose.

The discrepancies for nonsense and synonymous standards and for previously characterized variants is considerable. In the absence of a method that provides probability of pathogenicity for each variant, and allows clear assessment of each variant, it is not clear how the presented data can be utilized to inform the clinical classification of any CHEK2 variants.

The authors combine or merge SNVs within single amino acids by combining or averaging scores. By their own admittance, there are nonsense variants that yield tolerant activity and there are likely synonymous variants that yield damaging activity. There are also potential effects on RNA splicing (which is not measured or predicted here) and on RNA stability. For this reason the standard in the field is moving towards analysis of individual SNVs rather than amino acids. The authors should re-evaluate their data at the SNV level with this in mind.

Statements reporting “significantly more depleted” are not detailed enough. Numbers of variants with discrepancies should be shown.

Reporting of the correlations with VEPs, such as AlphaMissense, needs to include measures of sensitivity and specificity.

The report that the last 43 amino acids are not important for function of CHEK2 is interesting. However, the authors then show that 10.6% of variants in the C-terminus are damaging. This is confusing. Also, it is not clear if the C-terminus is considered in the reporting of missense variant effects. A clear explanation of the relevance of the terminal 43 amino acids for all aspects of the analysis is needed.

It is not clear how the authors account for possible temperature sensitivity effects on protein stability caused by individual variants in the yeast model.

The authors missed an opportunity to emphasize the clinical utility of their findings. Strangely, results from case-control associations studies for breast cancer are mentioned at the end of the Discussion but no details are provided in Results or Methods. This must be corrected.

It is absolutely essential that correct terminology be used throughout the manuscript. “Classification” is a term reserved for clinical classification or determination of the clinical relevance of a variant. Functional studies do not “classify” variants. Functional evaluation and “categorization” of variants should be used in this manuscript in the absence of any information on clinical relevance.

**Have all data underlying the figures and results presented in the manuscript been provided?**

Reviewer #1: **No: **It is not clear how the raw data from this report will be made available via a public repository.

Reviewer #2: Yes

Reviewer #3: None

Reviewer #4: Yes

PLOS authors have the option to publish the peer review history of their article (what does this mean?). If published, this will include your full peer review and any attached files.

Reviewer #1: No

Reviewer #2: No

Reviewer #3: No

Reviewer #4: No

---

## [Decision Letter · Decision Letter 1]

1 Jul 2024

Dear Dr Finley,

Thank you very much for submitting your Research Article entitled 'Comprehensive analysis of the functional impact of single nucleotide variants of human CHEK2' to PLOS Genetics.

The manuscript was fully evaluated at the editorial level and by independent peer reviewers. The reviewers appreciated the attention to an important topic but identified some minor concerns that we ask you address in a revised manuscript.

We therefore ask you to modify the manuscript according to the review recommendations. Your revisions should address the specific points made by each reviewer.

Yours sincerely,

Haico van Attikum

Guest Editor

PLOS Genetics

David Kwiatkowski

Section Editor

PLOS Genetics

Reviewer's Responses to Questions

**Comments to the Authors:**

Reviewer #1: Thank you for your careful and considered response to my comments, questions and suggestions.

Could the authors be more explicate about their data not supporting the use of intermediate functional category in the text of the manuscript? The spread of variants called intermediate in other assays across the RCS scale (as illustrated in the figure provided in the response), helps to further understand the nature of the data and why a third category is not proposed by the authors.

Reviewer #2: The authors have adequately addressed my comments. This manuscript will be of great interest to many researchers/clinicians in the field of oncogenetics.

Reviewer #3: I thank the authors for their careful consideration of my comments, and I now think this paper is ready for acceptance. However, I would suggest it is very important that the variant scores first be deposited in MaveDB so that the accession number can be included in the paper. You can do this and set your dataset to private, changing it to public after acceptance

**Have all data underlying the figures and results presented in the manuscript been provided?**

Reviewer #1: Yes

Reviewer #2: Yes

Reviewer #3: None

PLOS authors have the option to publish the peer review history of their article (what does this mean?). If published, this will include your full peer review and any attached files.

---

## [Editor Report · Decision Letter 2]

25 Jul 2024

Dear Dr Finley,

We are pleased to inform you that your manuscript entitled "Comprehensive analysis of the functional impact of single nucleotide variants of human CHEK2" has been editorially accepted for publication in PLOS Genetics. Congratulations!

Yours sincerely,

Haico van Attikum

Guest Editor

PLOS Genetics

David Kwiatkowski

Section Editor

PLOS Genetics

Comments from the reviewers (if applicable):

**Data Deposition**

http://datadryad.org/submit?journalID=pgenetics&manu=PGENETICS-D-24-00029R2

**Press Queries**

---

## [Editor Report · Acceptance letter]

12 Aug 2024

PGENETICS-D-24-00029R2 

Comprehensive analysis of the functional impact of single nucleotide variants of human CHEK2 

Dear Dr Finley Jr., 

We are pleased to inform you that your manuscript entitled "Comprehensive analysis of the functional impact of single nucleotide variants of human CHEK2" has been formally accepted for publication in PLOS Genetics! Your manuscript is now with our production department and you will be notified of the publication date in due course.

With kind regards,

Zsofia Freund

PLOS Genetics

On behalf of:
